# Analysis of the Local Health-Enhancing Physical Activity Policies on the French Riviera

**DOI:** 10.3390/ijerph18010156

**Published:** 2020-12-28

**Authors:** Antoine Noël Racine, Jean-Marie Garbarino, Bernard Massiera, Anne Vuillemin

**Affiliations:** Graduate School of Health Science Ecosystems, Université Côte d’Azur, Lamhess, 06200 Nice, France; antoine.noel-racine@etu.univ-cotedazur.fr (A.N.R.); jean-marie.garbarino@univ-cotedazur.fr (J.-M.G.); bernard.massiera@univ-cotedazur.fr (B.M.)

**Keywords:** health promotion, physical activity, local government, policy analysis

## Abstract

Policy is a lever for initiating the structural and environmental changes that foster health-enhancing physical activity (HEPA) promotion. However, little is known about the evidence in support of local governments regarding their HEPA-promoting policies. The aim of this study was to collect comprehensive information on municipal HEPA policies on the French Riviera (Alpes-Maritimes and Var counties) to provide an overview of the development of these policies in this territory. Mid-sized cities from the two counties constituting the French Riviera were targeted (*n* = 17). In each city, a local tool for HEPA policy analysis, CAPLA-Santé, was used to gain information from key informants heading the departments of sports, health services, and social services. Data were collected through semi-structured interviews and document analysis. Ten mid-sized cities volunteered to participate. Key informants from the sports (*n* = 10), health services (*n* = 5), and social services (*n* = 6) departments were interviewed. Written HEPA policy documents were formalized in six cities. These documents (*n* = 14) were mainly from the sports (*n* = 8) and health services (*n* = 4) sectors. The key informants reported that support from national policy, the commitment of elected officials, and large local stakeholder networks facilitated HEPA promotion, whereas the lack of intersectoral collaboration and limited resources were limitations. The results provide insight into the development of municipal HEPA policies, highlighting some of the barriers, facilitators, and perspectives. These findings could be valuable to scale up HEPA promotion at the local level.

## 1. Introduction

The health benefits of physical activity to prevent non-communicable diseases are well established [1,2]. Health-enhancing physical activity (HEPA) promotion is thus an important issue in the public health field [3,4]. Research has mostly focused on individual-level factors and showed that ecological models posit that the physical and social environments are important determinants of physical activity [5]. According to the literature, policy is one of the levers for initiating structural and environmental change to foster HEPA promotion [6,7,8,9]. The development of sustainable HEPA policies would provide people with a wide range of real opportunities to adopt physically active lifestyles [8,10,11,12]. It can be emphasized that the development of HEPA policies may be affected by the perceptions of the potential levers and barriers [13] and it has been shown that strong HEPA policy interventions have an impact on population health [8].

Nevertheless, the data on physical activity levels have shown that a significant part of the world’s population is insufficiently physically active [14]. In France, 31% of adults and 81% of young people do not meet the World Health Organization (WHO) recommendations for physical activity (150 min per week for adults and 60 min per day for young people) [15,16]. Since the early 2000s, successive French national governments have developed HEPA policies to address the problem of physical inactivity [17,18,19].

However, solving this problem is a complex undertaking [20], and the involvement of national or federal governments alone is insufficient [21]. Effective HEPA promotion also requires the involvement of local governments, including a wide range of sectors and stakeholders [6,21,22]. Local governments are particularly suitable for implementing intersectoral policies to promote HEPA as they can more easily influence the conditions and environments where people live [10,23,24]. This means, of course, that a local government has to deal with a complex ecosystem embedded in a specific context [25].

Yet little evidence has emerged in support of local governments regarding their policies to promote HEPA [26]. A recent review exploring the published research on local-government HEPA policies indeed showed that the scientific literature on the topic remains scarce [27]. The analysis of local HEPA policies would therefore likely shed light on the gaps and opportunities in the initiatives of policymakers and researchers to increase the physical activity levels of various populations [28]. More extensive monitoring of HEPA promotion efforts through policy indicators would undoubtedly improve policymaking and policy decisions [8]. A better understanding of how local HEPA policies are developed and implemented would help provide suitable solutions to local governments. The aim of this study was to collect comprehensive information on municipal HEPA policies on the French Riviera to provide an overview of the development of these policies in this territory.

## 2. Materials and Methods

### 2.1. Description of the CAPLA-Santé Tool

CAPLA-Santé (Cadre d’Analyse des Politiques Locales Activité physique-Santé—Analysis framework of Local HEPA Policies), version. company, city, country, a tool for analyzing local HEPA policies, was developed in collaboration with the French Society of Public Health [29]. It is based on the WHO HEPA Policy Audit Tool (PAT) version 2, company, city, country [28,30], which analyzes national HEPA policies. Briefly, a multidisciplinary and intersectoral group of experts, including researchers, professionals, and policymakers (national and local levels), was involved in developing CAPLA-Santé. The experts adapted each item of the HEPA PAT v2 to the local level (e.g., instead of asking leadership and collaboration at the national level, it was reformulated for the local level: Question 4 “Are there organizations or bodies which ensure cross-sectoral collaboration or coordination in implementing HEPA policies and action plans across the local government area studied?”). After a test within seven local governments to obtain feedback on the framework, a final workshop was organized to adjust and finalize it. The final version of CAPLA-Santé contains 21 items divided into six major sections: overview of HEPA stakeholders in the local government area, policy documents, policy contents, funding and political engagement, studies and measures related to physical activity in the local government area, and progress achieved and future challenges.

### 2.2. Participants

Mid-sized municipalities (between 20,000 and 100,000 residents according to the National Institute of statistics and economic studies (INSEE) [31]) from the French Riviera (Alpes-Maritimes and Var counties) were invited to participate in this study (*n* = 17). Mid-size municipalities were chosen to ensure more homogeneous municipalities; smaller ones have fewer resources to develop HEPA policies and there are only two big municipalities (over 100,000 inhabitants) with a different magnitude of resources compared with mid-size municipalities. Municipalities were initially contacted by e-mail. If necessary, a phone call was organized to provide more detail on the research project. The municipalities that volunteered to participate in the project were included in the study. Data on the characteristics of each municipality (number of inhabitants, median income per inhabitant, number of people affected by a chronic illness) were collected from the Regional Health Observatory of Provence-Alpes-Côte d’Azur database [32]. In each municipality, the objective was to recruit a minimum of two key informants from three main sectors: sports, health services, and social services.

### 2.3. Data Collection and Analysis

Data were collected between September 2018 and March 2019 from two sources: written HEPA policy documents collected from the internet (municipality websites) and key informants from the sports, health services, and social services sectors of the municipalities included in this study. The definition of a written HEPA policy document used in this study was the one used in CAPLA-Santé: “Written document that contains priorities, defines goals and objectives, and usually comes from a specific sector of public administration” [29]. To be reviewed, the HEPA policy documents had to have a clearly expressed health objective. The content of these documents was analyzed using the items and major sections of the CAPLA-Santé. Through semi-structured interviews, face to face or by phone, all items of the CAPLA-Santé were addressed to the key informants from the sports, services, and social services sectors to elicit their responses. Each interview was digitally recorded and transcribed verbatim. After data analysis, a final report and a synthesis of the CAPLA-Santé results were provided to each municipality. Data from each municipality were added to a matrix including all items of CAPLA-Santé for a global analysis. Ethical approval was obtained from Université Côte d’Azur before starting the study under the reference UCA-E19-011.

## 3. Results

### 3.1. Participants

Among the 17 invited mid-sized municipalities, 10 volunteered to participate in this study. In these municipalities, 21 key informants heading the departments of sports (*n* = 10), health services (*n* = 5), social services (*n* = 5), and the department of sports and social services (*n* = 1) were involved. Table 1 presents the characteristics of the municipalities and key informants interviewed.

### 3.2. Data Collection and Analysis

The average time of the semi-structured interviews was 35 min.

#### 3.2.1. Section 1: Overview of HEPA Stakeholders in the Local Government Area

The key informants from the municipalities identified an average of four public agencies engaged in HEPA promotion in their territory. These agencies were mainly from the sports (*n* = 15), health services (*n* = 13), and education (*n* = 6) sectors. Among the 10 municipalities included in this study, six considered themselves leaders in pushing forward HEPA promotion within their territory, three had a department head who ensured cross-sectoral collaboration or coordination in implementing the HEPA policies, and five had connections with a HEPA promotion network. Table 2 presents the section details for each municipality. 

#### 3.2.2. Section 2: Policy Documents

Fourteen written HEPA policy documents collected from six municipalities were reviewed (Figure 1). These policy documents were from the sports (*n* = 8), health services (*n* = 4), social services (*n* = 1), and environment (*n* = 1) sectors.

#### 3.2.3. Section 3: Policy Contents

Of the 14 written HEPA policy documents, none included quantitative objectives. The objectives expressed in these documents differed according to the sector and the target audience, such as “to improve the health of people suffering from chronic diseases through physical activity”, or “to facilitate access to free and outdoor physical activity in line with the public health recommendations” or “to maintain the independence of elderly people with physical activity.” Table 3 provides details on the contents of the policies for each municipality.

#### 3.2.4. Section 4: Funding and Political Engagement

Only the municipalities that had a HEPA policy (*n* = 6) identified funding sources to implement these policies. Most of the funds were provided by other government bodies at local and national levels, such as the Agence Régionale de santé (i.e., the Regional Health Agency), Agenda 21 (i.e., an environmental action plan to be implemented at local level), and the Conférence des Financeurs de la Prevention de la Perte d’Autonomie (i.e., the County Conference of Funders for the Prevention of Loss of Autonomy). Municipality “A” also identified funding from the private sector (insurance). No source of funding was recurring (1-year maximum).

#### 3.2.5. Section 5: Studies and Measures Relating to Physical Activity in the Local Government Area

No municipality reported a surveillance system related to physical activity. Municipality “A” conducted two surveys from a HEPA policy development perspective: one to explore the physical activity expectations and needs of the target audience and one to measure obesity in the schools. No municipality reported a cost-benefit study related to physical activity.

#### 3.2.6. Section 6: Progress Achieved and Future Challenges

Table 4 summarizes the key moments in the development of the municipal HEPA policies, the strengths and weaknesses in developing them, and the progress achieved and future challenges.

## 4. Discussion

The results provided an overview of HEPA policy development in mid-sized municipalities from the French Riviera. They also helped identify the stakeholders that should be involved in HEPA policymaking at the local level, the types of HEPA policies that mid-sized French municipalities are able to develop, and the factors that are likely to facilitate or limit their development. Last, the results pointed to perspectives on HEPA policy development that might be explored and provided future research and practice directions. For example, we showed that when a dense network of stakeholders was involved in HEPA promotion within the territory, the municipalities formalized more of the written policy documents. Similarly, most of the municipalities with written HEPA policy documents had connections with a HEPA network. Although this cross-data analysis indicated no cause–effect relationship, it suggested that HEPA promotion as an item on the municipal policy agenda might be facilitated by advocacy from a local HEPA network.

A theory often used to determine how an issue such as HEPA promotion finds a place on the policy agenda is the Multiple Streams Approach (MSA) [33]. According to Kingdon et al. (1995), the advocacy of a strong “policy entrepreneur” such as a stakeholder network can facilitate the opening of a “policy window” through which an issue finds a place on the policy agenda [21]. Using the MSA with data collected from the CAPLA-Santé tool in a range of contexts and with additional municipalities might well highlight the influence of local stakeholder networks on the policy agendas of municipalities in general. It would then be interesting to explore how strong local advocacy can stimulate and scale up the development of HEPA policies [34].

In addition, national advocacy might also stimulate and scale up the development of these local policies. The results from the section on policy documents showed that the municipalities that had the most and most varied HEPA policies had based their policies on national policy and/or research evidence. Although in line with the literature [6,22,35] and the WHO recommendations [11,12], there is still a need to better understand the influence between national and local HEPA policies and better translate research evidence to policymaking [8].

Another direction for investigation concerns the opportunities and difficulties that municipalities encounter using an environmental approach [6,7,36]. In this study, the municipalities rarely implemented concrete actions based on this approach, which is not in line with the literature [6,7,36]. Studies have shown that HEPA policies should prioritize environmental approaches, such as developing active transport, green parks, and open spaces, over individual approaches [7,8,10,36]. Moreover, doing so makes it easier to target a larger proportion of the population. The difficulty of adopting an environmental approach could be partially explained by a lack of intersectoral collaboration [37,38,39]. Our data showed that for the municipalities that had HEPA policies, few sectors were involved outside of sports and health services, with, for example, the urban and environment sectors rarely involved. Yet, it has been well documented that an intersectoral approach is needed to achieve an effective HEPA policy [6,8]. A dependable bridge from the latest research evidence to the policymakers may thus be crucial to ensure the effectiveness of HEPA policies [40].

A surveillance system is also important to ensure the provision of relevant information to guide the development, implementation, and adaptation of HEPA policies [8,41]. However, no municipality in this study had a surveillance system for PA or sedentary behavior. This type of surveillance should be based on repeated measures to provide data on PA and sedentary behaviors, as well as the associated factors [41]. Moreover, for municipalities that have a surveillance system, it is probably not a priority, given the frequent budget constraints and limited human resources for running it. Nevertheless, with the emergence of smartphone accelerometers and applications to measure PA [8,42], municipalities now have a great opportunity to set up a surveillance system with few resources.

This study had certain limitations. The HEPA policy analysis was limited to two counties in France. Moreover, only mid-sized municipalities were recruited in these counties. We assumed that smaller municipalities (fewer than 20,000 inhabitants according to INSEE [31]) would have even fewer resources to develop HEPA policies than mid-size municipalities, whereas big municipalities (more than 100,000 inhabitants according INSEE [29]) would have more but would be more complex to analyze. In addition, the magnitude of the resources may have differed across the mid-sized municipalities as it was difficult to assemble a homogeneous sample of municipalities. Similarly, it was difficult to recruit key informants who had enough time to participate in this study. Those that we interviewed were from only three sectors (sports, health services, and social services) and information from other sectors, such as the urban or environment sectors, may have been missed. Last, the data from the interviews may have been biased due to social and political desirability [43]. However, the strengths of this study should be emphasized. Few studies were interested in local HEPA policies in several municipalities. A standardized framework was used to collect data on HEPA policies which provide an overview and allow comparison. This study represents the first step in the implementation of a follow-up of these policies, using the CAPLA-Santé which aims to capture the progress and experiences on developing local HEPA policies.

## 5. Conclusions

This study collected comprehensive information on local HEPA policies on the French Riviera in order to obtain an overview of the HEPA policies within this territory. Our results may contribute to a better understanding of the development of local HEPA policies as they highlight some of the barriers, facilitators, and perspectives. The CAPLA-Santé tool was able to provide policy information that may be helpful to policymakers in their future HEPA policy decisions. It could help to enhance or strengthen the capacity-building of local governments as they develop HEPA policies. Nevertheless, municipalities may need to be more systematic about adopting intersectoral and environmental approaches to enhance the promotion of physical activity. Further studies in different contexts, including cities with a greater number of inhabitants, as well as reviews of HEPA policy development and implementation, and research translation into practice are now needed to help implement these approaches.

## Figures and Tables

**Figure 1 ijerph-18-00156-f001:**
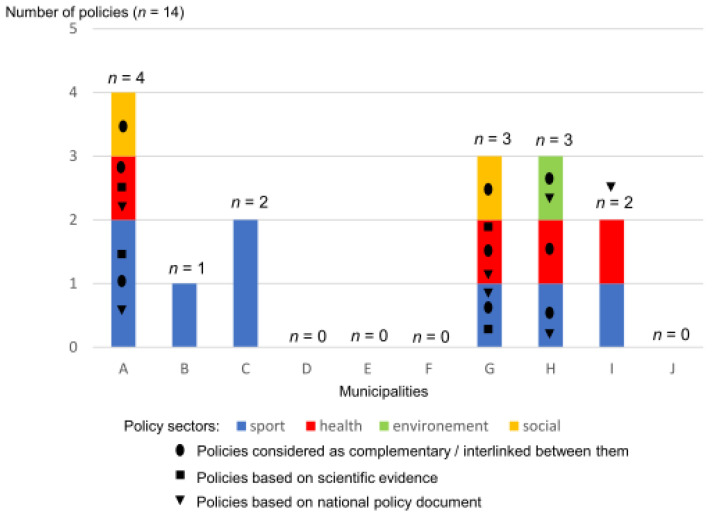
HEPA policy documents collected from municipalities.

**Table 1 ijerph-18-00156-t001:** Characteristics of the municipalities and key informants.

Municipality	Inhabitants (*n*) *	Median Income (€) **	People Affected by a Chronic Illness (*n*) ***	Key Informant Position	Sector Position
A	74,875	22,392	12,441	HD	social
				Head of HEPA † project Project officer	sport health
B	49,322	22,046	8012	HD	social
				HD	sport
C	28,919	22,858	4592	HD	social
				HD	sport
D	50,937	20,704	7607	HD	sport and social
E	41,571	20,010	7250	HD	sport
				HD	health
F	35,296	23,152	6913	HD	sport
				HD	health
G	64,903	18,656	11,305	HD	sport
				HD	health
H	74,285	18,962	14,369	HD HD	sport social
I	25,047	20,940	4656	HD	social
				HD	sport
J	23,347	21,778	3342	HD	social
				HD	sport
				HD	health

Note: Head of the department (HD) * Data from the National Institute of Statistics and Economic Studies—INSEE (2018). ** Data from INSEE (2018). *** Number of people affected by a chronic illness covered by government insurance for their healthcare expenditures. Data from the Regional Health Observatory of Provence-Alpes-Côte d’Azur (2018). † HEPA: health-enhancing physical activity

**Table 2 ijerph-18-00156-t002:** Overview of HEPA stakeholders in the local government area.

Municipality	Sectors and Public Agencies (*n*) Engaged in HEPA Promotion	Non-Governmental Stakeholders (*n*) Engaged in HEPA Promotion	Leadership Identified	Coordinator Identified	Connections with HEPA Network
A	sport (*n* = 2), health (*n* = 1), multisectoral (*n =* 1)	insurance and private sport (*n* = 2), sport (*n* = 1), health (*n* = 1)	municipality	sports department	yes
B	sport (*n* = 3), multisectoral (*n =* 2) health (*n* = 1), education (*n* = 1)	sport (*n* = 1), health (*n* = 1)	municipality	-	yes
C	sport (*n* = 2), health (*n* = 1), social (*n* = 1), multisectoral (*n* = 1)	sport (*n* = 3), health (*n* = 2), insurance and private sport (*n* = 2)	municipality	-	yes
D	health (*n* = 3), education (*n* 2), sport (*n* = 1),	sport (*n* = 1)	-	-	-
E	sport (*n* = 2), health (*n* = 1)	sport (*n* = 1), health (*n* = 1)	-	-	-
F	multisectoral (*n* = 1)	sport (*n* = 1), health (*n* = 1), insurance (*n* = 1)	-	-	none
G	health (*n* = 3), sport (*n* = 2), education (*n* = 1); multisectoral (*n* = 1)	health (*n* = 2), private sport (*n* = 1), sport (*n* = 1)	municipality	social services department	yes
H	sport (*n* =1), health (*n* = 1), education (*n* = 1)	sport (*n* = 1), health (*n* = 1)	municipality	Health services	yes
I	health (*n* = 2), sport (*n* = 1),	sport (*n* =1), health (*n* = 1), education (*n* = 1)	municipality	-	-
J	sport (*n* = 1), education (*n* = 1)	sport (*n* = 1), education (*n* = 1)	-	-	-

- none.

**Table 3 ijerph-18-00156-t003:** Policy contents.

Municipality	Policy Settings	Target Audiences	Communication Strategies or Actions	Concrete Actions
A	sports and leisure urban environment health and social care centers	general population seniors inactive and people suffering from chronic diseases	website social networks local newspapers events awareness of healthcare professional	PA * program fitness trails
B	sports and leisure	sedentary people	website events	PA program
C	sports and leisure	seniors	website social networks local newspapers	PA program
D	no data	no data	no data	no data
E	no data	no data	no data	no data
F	no data	no data	website social networks	events promoting PA for heath
H	sports and leisure tourism urban environment	general population inactive people pre-school and children	website social networks events	outdoor fitness and trail network HEPA events
I	sports and leisure primary school priority neighborhoods for urban policy	seniors children	website social networks	PA programs
J	no data	no data	no data	no data

* PA: physical activity.

**Table 4 ijerph-18-00156-t004:** Synthesis of Section 6: progress achieved and future challenges.

Municipality	Keys Moments	Strengths	Weaknesses	Progress	Challenges
A	national legislation of physical activity prescription; local conference on the topic	local stakeholder network; geographic situation; sportsfacilities; presence of PA and health professionals	limited resources; lack of cycle path network	implementation of HEPA actions	formalize global HEPA action; develop HEPA events; identify recurring funding to sustain HEPA policies
B	pilot implementation of PA program	local stakeholder network; quality and number of PA facilities	lack of intersectoral coordination; geographic difficulties in accessing PA facilities	sustainment of a PA pilot program for seniors to a regular program	improve stakeholder coordination; identify recurring findings to sustain HEPA policies; develop public space to practice PA
C	mayor’s willingness to promote PA	local stakeholder network; geographic situation	few PA programs available for sedentary and inactive people; lack of resources; overuse of PA facilities	implementation of HEPA actions	develop human resources with PA and health training; identify recurring funding to sustain HEPA policies
D	no data	local stakeholder network	unwillingness to develop policy; difficulties moving in the city without a car; lack of resources	no data	build willingness to develop a policy; develop a global intersectoral HEPA policy
E	no data	knowledge of the territory	lack of resources	no data	develop active mobility; develop a HEPA plan
F	no data	local stakeholder network; quality and number of sports facilities; knowledge of the local context	unwillingness to develop policy; lack of knowledge in PA and health, lack of intersectoral collaboration	no data	formalize a HEPA policy; develop a campaign to sensitize residents
G	-national policy to promote PA	local stakeholder network; culture of sport; global vision of health	lack of resources and PA facilities	implementation of HEPA actions	develop a global HEPA project from children to older people
H	-mandate of the mayor; national HEPA campaign	local stakeholder network, geographic situation; willingness of the mayor; PA facilities	overcrowed PA facilities	implementation of HEPA actions	target more inactive people; develop more cyclable paths
I	-national policy to promote PA	geographic situation; PA facilities; good communication	lack of resources	implementation of HEPA actions	target more inactive people; develop a global intersectoral HEPA policy with dedicated human resources
J	no data	local network of stakeholders	lack of PA facilities and lack of public open space	no data	build willingness to develop a policy; develop a global HEPA policy with the metropolis

## Data Availability

The data that support the findings of this study are available from the corresponding author, upon reasonable request.

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
