# Peer review of "Analysis of the Local Health-Enhancing Physical Activity Policies on the French Riviera"

_ijerph, 2020, doi:10.3390/ijerph18010156_

Round 1
Reviewer 1 Report
The manuscript entitled “Analysis of the local health-enhancing physical activity policies on the French Riviera” deals with a study of the implementation of health-enhancing physical activity (HEPA) amongst a certain region of France focusing on mid-size cities (20 000 – 100 000 inhabitants) in order to analyze the best approaches to increase physical activity (PA) in the cities. The authors applied an already-made framework for analyzing local HEPA policies, finding various reasons why HEPA is not sufficiently applied in the cities, but also finding strategies for achieving this goal.
- In my opinion, the article addresses a worldwide interesting topic. Nevertheless, the results have a certain limitation since only are focused on the public PA and how the local governments have to deal with it. What is the opinion of the authors regarding other types of PA (indoor, for instance) and how does HEPA include it on its agenda?
- Results of the analysis are sort-of predictable: the main challenges among the municipalities are the lack of economic resources and organization amongst the several stakeholders. What other solutions can be given to increase the HEPA programs, considering other kind of PA such as particular indoor exercise and private PA programs?
- Apparently, a key factor has been forgotten within the manuscript: the surveys were headed to the head of the local departments of health services, social services etc., but not headed to the local population. What is the opinion of the respective local population? Was it taken into account? Each city must have its own challenge regarding the HEPA implementation. How the population is involved to consider the best solution for each challenge in particular? On the manuscript, this subject is not clearly presented.
Specific comments:
Lines 11-13: This sentence is redundant. Please correct it.
Line 13: Which counties?
Line 38: Please summarize the recommendations of WHO.
Line 61: State that “Framework for Analyzing Local HEPA Policies” is the translation into English of “Cadre d’Analyse des Politiques Locales Activité physique-Santé”.
Line 66: Authors claim that CAPLA-Santé has applied HEPA at local level. Nonetheless, at line 50 they claim that literature on local-government HEPA is scarce. Please rephrase the sentences to avoid confusion on the reader.
Line 98: Please explain why the number of departments does not match with the number of key informants.
Author Response
The manuscript entitled “Analysis of the local health-enhancing physical activity policies on the French Riviera” deals with a study of the implementation of health-enhancing physical activity (HEPA) amongst a certain region of France focusing on mid-size cities (20 000 – 100 000 inhabitants) in order to analyze the best approaches to increase physical activity (PA) in the cities. The authors applied an already-made framework for analyzing local HEPA policies, finding various reasons why HEPA is not sufficiently applied in the cities, but also finding strategies for achieving this goal.
In my opinion, the article addresses a worldwide interesting topic. Nevertheless, the results have a certain limitation since only are focused on the public PA and how the local governments have to deal with it. What is the opinion of the authors regarding other types of PA (indoor, for instance) and how does HEPA include it on its agenda?
A recent scoping review showed that the literature remained scant about local public HEPA policies (Noël Racine et al., IJSPP 2020). Based on this scoping review the authors focused this article on local public HEPA policies from municipalities of the French Riviera. HEPA policies acting on every kind of PA (including indoor) were collected. Other PA policies exist but developed by sport federations or other bodies which are not directly linked to PA policies developed by local governments. It was a choice to only focused the study on these policies. Moreover, physical and sports activity in France has historically been attached to local communities. For this reason, the authors have not developed further on other types of PA and how does HEPA include it on its agenda.
Results of the analysis are sort-of predictable: the main challenges among the municipalities are the lack of economic resources and organization amongst the several stakeholders. What other solutions can be given to increase the HEPA programs, considering other kind of PA such as particular indoor exercise and private PA programs?
This was not the main results of the analysis, but part of the data collected. The lack of resources reflected the perception of local governments. This could be probably overcome with more intersectoral collaboration including with the private sector.
Apparently, a key factor has been forgotten within the manuscript: the surveys were headed to the head of the local departments of health services, social services etc., but not headed to the local population. What is the opinion of the respective local population? Was it taken into account? Each city must have its own challenge regarding the HEPA implementation. How the population is involved to consider the best solution for each challenge in particular? On the manuscript, this subject is not clearly presented.
We followed the same methodology used with the HEPA-PAT V2 of the World Health Organization, so the survey was addressed to key informant who develop and implement the HEPA policies. However, the CAPLA-Santé investigate if local governments make survey headed to their local population and the results were presented in 3.2.5. Section 5: Studies and measures relating to physical activity in the local government area.
Specific comments:
Lines 11-13: This sentence is redundant. Please correct it.
The sentence has been corrected
Line 13: Which counties?
Counties has been detailed
Line 38: Please summarize the recommendations of WHO
Recommendations has been summarized
Line 61: State that “Framework for Analyzing Local HEPA Policies” is the translation into English of “Cadre d’Analyse des Politiques Locales Activité physique-Santé”.
Yes, the authors confirm that it is the translation of French into English. The translation has been modified “Analysis framework of Local HEPA Policies”.
Line 66: Authors claim that CAPLA-Santé has applied HEPA at local level. Nonetheless, at line 50 they claim that literature on local-government HEPA is scarce. Please rephrase the sentences to avoid confusion on the reader.
The authors don’t understand the modification to provide. The literature on local-government HEPA is scarce and explain why we developed the CAPLA-Santé (Noël Racine et al. Health Promotion Practice 2020). The first step was to develop the CAPLA-Santé and after, we used it in this study.
Line 98: Please explain why the number of departments does not match with the number of key informants
Thank you for this comment. One department was missing: “Among 17 invited mid-sized municipalities, 10 volunteered to participate in this study. In these municipalities, 21 key informants heading the departments of sports (n = 10), health services (n = 5) and social services (n = 5), the department of sports and social services (n = 1) were involved. Table 1 presents the characteristics of the municipalities and key informants interviewed.”

Reviewer 2 Report
Dear authors
the paper entitled "Analysis of the local health-enhancing physical activity policies on the French Riviera" addresses the issue of the policies that the institutions (the municipalities in this case) have set for the promotion of physical activity.
This is a topic that is not easy to explore, as the variables from area to area are treated differently even within the same country. This is indeed a critical point to be explored but also to bring to the attention of the institutions.
The manuscript is well written with a simple descriptive study design.
- In the discussion I suggest to put first the limits of the study and then the future directions as a new direction of study could be to analyze the cities with a greater number of inhabitants.
- I suggest to include the strengths of the study.
Author Response
In the discussion I suggest to put first the limits of the study and then the future directions as a new direction of study could be to analyze the cities with a greater number of inhabitants.
This idea has been added at the end of the conclusion paragraph “Further studies in different contexts, including cities with a greater number of inhabitants, as well as review of HEPA policy development and implementation, and research translation into practice are now needed to help to implement these approaches.”
I suggest to include the strengths of the study.
The strengths of the study have been added after the limitations: “However, the strengths of this study should be emphasized. Few studies were interested in local HEPA policies in several municipalities. A standardized framework was used to collect data on HEPA policies which provide an overview and allow comparison. This study represents the first step in the implementation of a follow-up of these policies, using the CAPLA-Santé which aims to capture the progress and experiences on developing local HEPA policies.”
Reviewer 3 Report
Thank you for the opportunity to review this manuscript. The authors examine especially the development of municipal policies in encouraging physical activity. Lack of physical activity is a major risk factor for development of diseases such as T2D, and thus the topic is relevant and important. The manuscript appears to describe a cross sectional descriptive investigation of publicly available policies with qualitative interviews.
In its current form the manuscript is difficult to understand as the authors appear to have left out a considerable amount of information that would have helped in following especially the methods and results sections. In addition, considering the limited amount of evidence the authors presented, the conclusions appear very definitive.
Considering that the authors have interviewed/spoken to some of stakeholders, it should be clarified in the text whether ethical approval was sought and when not, why. More detailed feedback is included below.
Abstract
It is unclear what the main aim of the research is. Authors mention support regarding HEPA-policies, but it is not clear whether this relates to policy development, implementation, or impact analysis.
Line 20: The… - This sentence is difficult to understand.
It is not clear from the abstract how aims, results, and conclusions fit together. This should be clearly formulated.
Introduction
The introduction lacks considerably in Depth, and should consider including information such as what are barriers to HEPA and how local policy development would be beneficial for development of HEPA behaviours. Authors should also expand on why policy development is a complex process, and e.g. why local governments can better influence complex local ecosystems national governments.
Does literature exist about how effective policy decisions are in increasing HEPA behaviours? Or if the literature is very limited, has policy making been successful in influencing other health behaviours such and consumption of “junk” food.
In addition, authors should reflect any available literature regarding HEPA behaviours that policy development can influence.
Research aims are also not clear, and it is difficult to get the sense what the authors are trying to achieve. Authors state that they aim to examine gaps in policy making, but it is not clear from the introduction what is expected from the local authorities, i.e. are the guidelines / strategy / legal requirements what local councils should be following / aiming for. Further, it is stated that authors want to investigate development and implementation of policy.
Overall, literature in introduction should reflect better both research aims.
From line 53 – this sentence is somewhat confusing, while appearing as a literature reference, it reads like ta research aims.
Methods
Line 66: Please could you provide an example of an adaptation to local level.
Please could you expand the information especially regarding your sampling methods. Regarding the selected municipalities; could you expand why mid-sized municipalities were selected, how many municipalities the two regions contained and what were the criteria that were used to decide whether a municipality was asked to participate of not. Finally, please include information about how it was decided that 17 municipalities would be invited. Please could you explain what the abbreviation INSEE stands for.
Line 75: From this point onward, the text discusses data collection procedures.
In addition to mentioning the data matrix, please could you add further information about data extraction and analysis methods. Policy document analysis / Analysis of information from e.g. head of departments.
Results
Please could you include shortly any reasons why municipalities declined participation.
Information provided in the table 1 about characteristics of the municipalities is limited. While there appears to be wish to protect the identities of the municipalities, based on exact population figures this should be easy to find out. Understanding the community characteristics would better enable to understanding the complex challenges faced by policy developers.
I am sorry, but the point of the section 3.2. escapes me.
It should be made clearer which tables include information from literature analysis and which from document analysis.
Discussion
First sentence – Evidence on which this was based on is not clear – how the results support this conclusion? Similarly, how did the results help in identifying which stakeholders should be included in the policy development – instead of who were already included? Was this a research question?
Overall, the selected methodology and a number of documents found appears to limit the authors ability to draw firm conclusions from this research. Therefore, while authors recognize the limitations for the research, the conclusions drawn are rather strong in comparison to the available evidence.
End of line 181 – reference please
Some of issues such as Multiple Streams Approach (MSA) and environmental should have been discussed in the introduction.
Author Response
In its current form the manuscript is difficult to understand as the authors appear to have left out a considerable amount of information that would have helped in following especially the methods and results sections.
In addition, considering the limited amount of evidence the authors presented, the conclusions appear very definitive.
The conclusions have been rephrased to be less assertive.
Considering that the authors have interviewed/spoken to some of stakeholders, it should be clarified in the text whether ethical approval was sought and when not, why.
Thank you for to remind the authors that they need to precise this ethical aspect. The following sentence has been added in the paragraph 2.3. Data collection and analysis: “Ethical approval was obtained from Université Côte d’Azur before starting the study under the reference UCA-E19-011.”
More detailed feedback is included below.
Abstract
It is unclear what the main aim of the research is. Authors mention support regarding HEPA-policies, but it is not clear whether this relates to policy development, implementation, or impact analysis.
The objective has been reformulated.
Line 20: The… - This sentence is difficult to understand.
There was a mistake in the sentence, the word “that” has been deleted. Thank you for pointing out this incomprehension.
It is not clear from the abstract how aims, results, and conclusions fit together. This should be clearly formulated.
The authors don’t understand the comment. It is difficult to give details on each section in the abstract. The aim was to collect information on municipal HEPA policies. The methods were presented (Data were collected through semi-structured interviews and document analysis.) and the results related to who were interviewed and if document were formalized.
Introduction
The introduction lacks considerably in Depth, and should consider including information such as what are barriers to HEPA and how local policy development would be beneficial for development of HEPA behaviours. Authors should also expand on why policy development is a complex process, and e.g. why local governments can better influence complex local ecosystems national governments.
Two references have been added on the determinants of physical activity (Bauman et al. Lancet 20212) and perceptions of barriers and levers of HEPA policies (Noël Racine et al. Health Research Policy and Systems 2020), in the introduction.
Does literature exist about how effective policy decisions are in increasing HEPA behaviours? Or if the literature is very limited, has policy making been successful in influencing other health behaviours such and consumption of “junk” food.
In addition, authors should reflect any available literature regarding HEPA behaviours that policy development can influence.
Research aims are also not clear, and it is difficult to get the sense what the authors are trying to achieve. Authors state that they aim to examine gaps in policy making, but it is not clear from the introduction what is expected from the local authorities, i.e. are the guidelines / strategy / legal requirements what local councils should be following / aiming for. Further, it is stated that authors want to investigate development and implementation of policy.
The aim of the study is presented in the abstract and at the end of the introduction. “The aim of this study was to collect comprehensive information on municipal HEPA policies on the French Riviera to provide an overview of the development of these policies in this territory.”
It has not been stated that the aim was to investigate development and implementation of policy. This work is a first step to implement a follow-up of these policies. A sentence has been added at the end of the Discussion paragraph: “This study represents the first step in the implementation of a follow-up of these policies, using the CAPLA-Santé which aims to capture the progress and experiences on developing local HEPA policies.”
And a sentence has been added in the Conclusions paragraph to provide perspectives related to development and implementation of HEPA policies: “Further studies in different contexts, including cities with a greater number of inhabitants, as well as review of HEPA policy development and implementation…”
Overall, literature in introduction should reflect better both research aims.
This study had only one aim. Because of the lack of literature on this topic the first aim was to have an overview of HEPA policies reported by local governments in the French Riviera.
From line 53 – this sentence is somewhat confusing, while appearing as a literature reference, it reads like ta research aims.
The research aim is stated lines 68-70. The previous sentences are introducing the aim, using references. For example, the lack of policy indicators was highlighted in the literature and the use of a framework, as done in this study using CAPLA-Santé, to collect data on HEPA policies may help to provide such indicators. But the first step is to give an overview of the HEPA policies using CAPLA-Santé. The next step will be to analyze more deeply how the policies were implemented.
Methods
Line 66: Please could you provide an example of an adaptation to local level.
An example has been added in 2.1. Description of the CAPLA-Santé tool: lines 80-83 “(e.g.. instead of asking leadership and collaboration at the national level, it was reformulated for the local level: Question 4 “Are there organizations or bodies which ensure cross-sectoral collaboration or coordination in implementing HEPA policies and action plans across the local government area studied?”)
Please could you expand the information especially regarding your sampling methods. Regarding the selected municipalities; could you expand why mid-sized municipalities were selected, how many municipalities the two regions contained and what were the criteria that were used to decide whether a municipality was asked to participate of not. Finally, please include information about how it was decided that 17 municipalities would be invited.
The 17 municipalities represent the total of eligible mid-size municipalities in the 2 counties (Alpes-Maritimes and Var). Smaller municipalities (under 20,000 inhabitants) have less resources to develop HEPA policies compared to mid-size municipalities. Thus, policymakers might not have the same experience and perception to HEPA policy development. In these counties, there are only two big municipalities (over 100,000 inhabitants) with a different magnitude of resources compared with mid-size municipalities. So, to ensure more homogeneous municipalities, the research team decided to select only mid-size municipalities. A sentence has been added in the section 2.2. Participants
Please could you explain what the abbreviation INSEE stands for.
The abbreviation INSEE means the National Institute of statistics and economic studies (INSEE). It has been clarified in the manuscript lines 90-91.
Line 75: From this point onward, the text discusses data collection procedures.
In addition to mentioning the data matrix, please could you add further information about data extraction and analysis methods. Policy document analysis / Analysis of information from e.g. head of departments.
A sentence has been added in 2.3. Data collection and analysis to explain that the content analysis of the policy documents followed the ietms and section of the CAPLA-Santé.
Results
Please could you include shortly any reasons why municipalities declined participation.
Information provided in the table 1 about characteristics of the municipalities is limited. While there appears to be wish to protect the identities of the municipalities, based on exact population figures this should be easy to find out. Understanding the community characteristics would better enable to understanding the complex challenges faced by policy developers.
The authors agree that having more information to understand the community characteristics would be interesting. However, asking more details was not included in the ethical approval obtained from the university.
I am sorry, but the point of the section 3.2. escapes me.
It should be made clearer which tables include information from literature analysis and which from document analysis.
The authors are not sure to understand the comment. The tables and figures presented in section 3.2 are the results of the interview-administered CAPLA-Santé. The results presented are not coming from the literature.
Discussion
First sentence – Evidence on which this was based on is not clear – how the results support this conclusion? Similarly, how did the results help in identifying which stakeholders should be included in the policy development – instead of who were already included? Was this a research question?
The first sentence has been moved to the conclusions. The authors understand the comment and it would be interesting to investigate further. However, as a first step, the aim of the study was to provide and overview of the situation in the French Riviera, not to explain what is done which will be a next step implying other investigations and methods.
Overall, the selected methodology and a number of documents found appears to limit the authors ability to draw firm conclusions from this research. Therefore, while authors recognize the limitations for the research, the conclusions drawn are rather strong in comparison to the available evidence.
The conclusions have been rephrased to be less assertive.
End of line 181 – reference please
References have been added.
Some of issues such as Multiple Streams Approach (MSA) and environmental should have been discussed in the introduction.
The authors made the choice to keep these points in the discussion because the study was not based on these models but were identified as interesting models to apply in further studies and were considered as perspectives.

Round 2
Reviewer 1 Report
The responses to the comments are very difficult to follow within the modified manuscript.
I presume the highlighted text on green addresses the comments but I'm not sure.
Reviewer 3 Report
Thank you for the opportunity to evaluate the manuscript again. The authors have considered the feedback and either made changes to improve the MS or explained why they do not wish to make changes. There appears to be some some spelling/grammar issues, but, based on the author responses, there are no further comments.